# A ddRADseq Survey of the Genetic Diversity of Rye (*Secale cereale* L.) Landraces from the Western Alps Reveals the Progressive Reduction of the Local Gene Pool

**DOI:** 10.3390/plants10112415

**Published:** 2021-11-09

**Authors:** Martino Adamo, Massimo Blandino, Luca Capo, Simone Ravetto Enri, Anna Fusconi, Michele Lonati, Marco Mucciarelli

**Affiliations:** 1Dipartimento di Scienze della Vita e Biologia dei Sistemi (DBIOS), Università degli Studi di Torino, Viale P.A. Mattioli 25, 10125 Torino, Italy; martino.adamo@unito.it (M.A.); anna.fusconi@unito.it (A.F.); 2Dipartimento di Scienze Agrarie, Forestali e Alimentari (DISAFA), Università degli Studi di Torino, Largo Paolo Braccini 2, 10095 Grugliasco, Italy; massimo.blandino@unito.it (M.B.); luca.capo@unito.it (L.C.); simone.ravettoenri@unito.it (S.R.E.); michele.lonati@unito.it (M.L.)

**Keywords:** crop genetic diversity, ddRAD sequencing, landrace, population structure, *Secale cereale*, Western Alps, rye

## Abstract

Rye (*Secale cereale* L.) has been at the basis of agriculture for centuries in most mountainous and northern areas of Eurasia, because it is more resistant than other cereals to low temperatures and poor soils. Rye deserves to be re-evaluated as a source of “environmentally resilient” genes in the future as well, and particularly in a perspective to grow cereals able to withstand global warming. According to recent studies, modern rye varieties have a relatively narrow genetic pool, a condition that is worsening in the most recent breeding processes. The preservation of local landraces as unique sources of genetic diversity has therefore become important, in order to preserve the genetic heritage of rye. In this study, genetic diversity of rye landraces collected in a sector of the Italian Alps particularly suited to traditional agriculture was investigated using the ddRADseq technique. A few landraces still managed with family farming turned out to be genetically distant from the commercial varieties currently in use, highlighting that the phenomenon of homogenization of the local genetic pool can be still circumvented. Ex situ conservation of genetically divergent landraces is a valid tool to avoid the dissipation of an as yet unexplored genetic potential.

## 1. Introduction

Globally, there are approximately 382,000 species of vascular plants on the Earth [1] and more than 6000 have been cultivated for food production [2]. However, fewer than 200 species have significant productions globally, with only nine (sugar cane, maize, rice, wheat, potatoes, soybeans, oil-palm fruit, sugar beet, and cassava) accounting for over 66 percent of all crop production by weight [2]. Between the 200 most cultivated crops we can individuate several thousands of varieties and landraces. A landrace is a domesticated, locally adapted, traditional variety that has developed over time, through adaptation to its natural and cultural environment, and due to isolation from other populations of the species [3,4].

Even if the genetic diversity within a crop can be broad, genetic erosion due to loss of variation and genetic vulnerability at the species level are growing matters of concern in several crops. In common rye (*Secale cereale* L.), as for other crops, the replacement of local landraces with modern varieties is one of the main reason for the loss of genetic variation.

Recent studies have shown that currently cultivated rye, especially modern cultivars, are genetically less diverse and differentiated than local landraces [5,6]. To obtain modern breeds, in fact, parental inbred lines are developed by recurrent selfing of parental plants [7,8]. In rye, narrowing of the genetic base impacts variation of different traits: variability for spring vs. winter habit, winter-hardiness, plant height, lodging resistance, disease resistance, seed color, straw stiffness, earliness, pathogen resistance, ploidy, and grain yield and quality [9,10]. The consequence is a decrease in selection gain and an increase in susceptibility to biotic and abiotic stresses, coupled with the threat of further genetic erosion [11].

According to a 2020 world survey genetic resources of rye include 26,100 seed accessions encompassing most of the species in the genus *Secale*, but which are much less than in wheat (555,449 accessions) and barley (339,563 accessions) [12]. Rye ex situ collections are lower than other cereals also because most of the species in the genus *Secale* are open pollinated and therefore much more difficult to maintain via regeneration of seed accessions.

Landraces are genetically heterogeneous populations generated in subsistence agriculture and, in this regard, they share with wild rye ancestors much more variation than modern varieties [12]. Thanks to the maintenance in situ by farmers, landraces are regarded as key components of plant biodiversity in world agro-systems, also under a cultural and social perspective. While before the 30s of the twentieth century, each mountain community cultivated its own crops, currently cultivated rye belongs to few commercial varieties [4]. During the “green revolution” in the 60s, many of these cultivars were abandoned for the more productive commercial varieties. Annual agricultural production benefited from this revolution at the expense of crop biodiversity and genetic variability [6,11]. These phenomena are more severe on mountains where depopulation forced a substantial change of patterns in the agro-pastoral land use at high-elevations. These coupled phenomena strongly reduced the number of already available local rye varieties (or landraces) [13].

In recent years, several initiatives have started to re-evaluate the importance of crop local varieties and to preserve their diversity by ex-situ and on-farm conservation, also in the perspective of feeding 50 billion people in 2050 [14]. Rye landraces have very rich and complex ancestry holding variation in response to many diverse stresses (low temperatures, drought, and low soil fertility) and are useful resources for the development of future crops deriving many sustainable traits from their heritage [15].

In Europe, for its use as food, feed and energy in biogas plants, rye is widely cultivated in Eastern, Central and Northern Europe, particularly in Poland, Germany, Ukraine and Russia. Even if subject to the ergot disease (*Claviceps purpurea* (Fr.) Tul.), rye presents several disease resistance genes reducing the need for phytochemicals [16,17]. Moreover, rye offers many traits for improved nutrition in cereals, particularly a high level of dietary fiber and a whole suite of minerals (Zn, Fe, P), arabinoxylans, beta-glucans, resistant starch, and bioactive compounds [10,18], with positive effects on human health [19]. Grain content of antioxidants, phenolics, and dietary fibers in rye landraces are presently under study (M. Blandino *pers. comm.*).

Recently, a complete genome of rye was assembled [20]; nevertheless, know-how on rye genetics still remains marginal, when compared to other important crops and cereals [21]. Several studies on relationships within [22,23,24] and between *Secale* species [25,26,27] have been carried out to study the genetic variability, using a variety of different methods. The molecular tools employed in these analyses include PCR-RFLP [28], RAPD [22,23], ISSR [23], AFLP [29,30,31], SAMPL [23], DArT [5,32], isoenzymatic markers [22,24], SSR [6,23,25,33,34], and single nucleotide polymorphisms (SNP) [35].

In this study, ddRAD-seq has been applied to examine the genetic diversity and structure within different *S. cereale* landraces sampled in a mountain range spanning from the Ligurian (Cuneo Province) to Pennine Alps (Verbano-Cusio-Ossola Province) in NW-Italy. In this area, rye was traditionally the most cultivated cereal for the production of flour [36,37] and of straw of good quality [38]. Less than one century ago, in Cuneo province alone, more than 28,000 ha were seeded with rye; now, the currently cultivated surface in whole Italy is 3580 ha (ISTAT 2020). By means of several interviews conducted among local populations, a few viable local rye varieties were tracked down. Landraces were also genetically compared with commercial ryes representative of EU market and with the perennial *Secale strictum* (Presl.) Presl. (syn. *Secale montanum* Guss.).

The aim of this study was to explore the genetic diversity in still-viable Italian NW-Alps rye landraces, in order to understand which of them deserve conservation efforts, due to their “authentic” adaptation to local environmental conditions. The most promising candidates will be kept as repositories of potential genetic diversity.

## 2. Results

We obtained 482,236,822 demultiplexed sequences representing 30,835 polymorphic loci among 31,595 genotyped loci. Globally we recorded 173,451 polymorphic sites. After clone-correction and informative loci filtering we obtained 17,705 loci and 119,814 polymorphic sites.

AMOVA showed no genetic variation among samples within populations (*p* > 0.05). This is confirmed by the high differentiation (Phi = 0.30; *p* < 0.001) between landraces/varieties. The overall variation within individuals is significant (Phi = 0.24; *p* < 0.001), revealing the presence of distinctive samples significantly divergent from the other individuals, and thus the presence of an inner genetic structure (Table 1).

Pairwise comparison of mean genetic distances between populations and species calculated by means of Fst statistics are reported in Figure 1. The heatmap of Figure 1a reports pairwise comparisons on the whole genetic dataset and shows that *Secale strictum* is genetically distinguished from all the remaining rye populations (mean Fst > 0.55); nevertheless, a group of four landraces grouped together at an intermediate distance between *S. strictum* samples and all the other landraces and varieties (Figure 1a; ARN, MIN, EST and SUN, see also the comment of Figure 1b). When pairwise comparisons were conducted only on *S. cereale* accessions, thus excluding *S. strictum* samples, we observed no other clear structures (Figure 1b). However, ARN, EST, and MIN landraces, now, showed significant moderate genetic distances from remaining rye accessions and high to very high genetic separation from the commercial hybrid Su Nasri (SUN) (Figure 1b). Moreover, ARN and MIN are genetically closer to EST and MAR than to any other accession. Very low genetic distances were registered between landrace pairs CAR and PEV, ROB and GAR, and DID and BRU, other than between the commercial open pollinated variety Antoninskie (ANT) and FRA.

Parameters of genetic diversity for rye accessions are presented in Table 2. The observed heterozygosity (Ho) ranged from 0.0281 (EST) to 0.0018 (GAR) and expected heterozygosity (He) from 0.00315 (EST) to 0.00185 (CAR), with an average of 0.00215 and 0.00218, respectively. In Table 2, rye accessions possessing negative F values, which corresponds to observed heterozygosity values greater than expected according to the Hardy-Weinberg equilibrium, are shaded in grey. F (inbreeding coefficient) ranged from 0.1079 (EST) to −0.2593 in SUN (the hybrid variety) with an average value of 0.003 accounting for a very low level of inbreeding and a high proportion of heterozygous individuals in almost all rye accessions. The nucleotide diversity (π) ranged from 0.00333 (EST) to 0.00146 (MUS) with an average of 0.00231 at the population level, and with no significant differences in absolute values with the two modern varieties.

Analysis of the genetic structure revealed the existence of three major genetic groups within the Alpine landraces (modal ΔK = 3) (Figure 2 and Appendix A; Appendix A). The larger group (cluster I) contained most of the rye landraces and the two modern varieties, SUN and ANT; the four landraces ARN, EST, MIN, and MAR grouped apart (cluster II) (Appendix A).

Estimated nucleotide diversity values (π) between populations rounded around the average value of 0.00231 (±0.00031 SD) (Appendix A). We observed two major deviations: the EST population with the bigger π of 0.00333 and the smaller value of 0.00146 for the MUS population. All the other populations own a similar nucleotide diversity; otherwise, results pointed out that generally populations of the cluster II (ARN, MAR, MIN, and EST) show values greater than commercial varieties SUN and ANT (Table 2; Figure 2a).

Results of the DAPC analysis helped to better understand genetic relationships between varieties and alpine landraces included in the large group of rye accessions (group I in Appendix A; Figure 3). The optimal number of principal components was 80, and we performed the analysis using 13 discriminant factors (nPop-1), as suggested by Jombart [41]. The multivariate analysis revealed how most landraces are tightly related to ANT, the commercial variety of rye, while the commercial hybrid Su Nasri (SUN) was relatively isolated. Nonetheless, some landraces were positioned relatively distant from ANT, such as DID and BRU (from the Cottian Alps in Turin Province), or MAC and ORM, which, on the contrary, share no clear relationships and are placed at the northernmost and southernmost position of the geographic distribution, respectively (Figure 2a).

When we further inspected relationships within this group of accessions by means of their Prevosti’s genetic distances (Appendix A), alpine landraces clustered coherently with results of the DAPC analysis, with the exception of the ORM landrace which was included in the main cluster with ANT. Setting the 0.02 genetic distance threshold among the ANT node, a big part of the rye landraces falls into the “ANT-like group”, revealing a larger genetic similarity within these landraces and the variety Antoninskie (ANT). Considering that all these populations were part of the K2 genetic cluster, MAC, DID, and BRU can be regarded as only marginally related to ANT. Moreover, landraces DID and BRU can be considered as a single genetic unit, and the same can be said for the pairs PEV-CAR, GAR-ROB-(MUS), and FRA-ANT (see also pairwise genetic distances in Figure 1b).

The contribution of each genetic locus employed to analyse genetic distances and differentiation between populations of the alpine landraces was different according to the explained variance in the DAPC multivariate analysis. The most informative loci determining genetic distances among the two clusters of landraces were 39; we manually inspected their functional annotation after a blast and eight of them resulted in known plant functions (Table 3).

## 3. Discussion

Genetic diversity of rye landraces was investigated in seed accessions derived from living populations sampled along the Italian Western Alps. We used an original approach for rye, ddRAD sequencing, which allowed us to sample genetic diversity for a large territory, where rye has been the crop key to the local economy (i.e., the Occitan Valleys of Piedmont) and an important source of livelihood, and nowadays, it has become a relict crop [36,38], historically funded on the cultivation of this ancient crop [37,49].

Seed accessions were obtained from local farmers, institutions (EAPAM) and from two seed banks (IPK and IAR). Landraces were genetically compared with two commercial ryes, largely marketed throughout Europe. The ddRAD sequencing allowed the analysis of 17,705 loci, an unprecedented deep analysis in rye, where several recent studies do not exceed the hundreds of loci analyzed [5,12,35].

### 3.1. Genetic Diversity

Traditional rye varieties are panmictic populations, characterized by high levels of heterozygosity and heterogeneity (seven in Bolibok-Brągoszewska et al. [5]). Previous observations conducted on SNP polymorphisms of wild, feral, and cultivated rye sampled in different parts of the globe had shown very small differences in total genetic diversity among geographical populations [35]. Conversely, in our study, the AMOVA showed that the majority of genetic diversity was present between populations (landraces and varieties) of the alpine rye, and individuals were globally significantly diverse among them. Moreover, our AMOVA showed very little genetic diversity between individuals of single populations, and their differentiation was not significant.

Hangenlad et al. [35] have explained the low differentiation found between Eurasian rye landraces as a consequence of the very low level of inbreeding (low F values), as expected by an outcrossing species. In a consistent group of landraces of our study, F values were significantly higher than in the work of Hangenlad et al. [35]. In 11 out of 16 alpine landraces, F values were positive and ranged from 0.03 to 0.10, thus indicating that the frequency of observed heterozygosity was lower than the Hardy-Weinberg expectation. In three landraces (MAR, EXI and EST), particularly, inbreeding coefficients exceeded 0.09, which is largely higher than mean F values found in literature (Hangenlad’s landraces; F = −0.116). In the case of the EST, an F value of 0.108 can be regarded as a serious indication of inbreeding. EST comes from a local farmer which cultivates rye seeds inherited by its father and no other farmers grow rye in the surroundings; this is also the case of MAR (F = 0.089; Table 2); F values are probably the result of self-pollination, repeated year by year.

Self-incompatibility in rye is not strict and, with the exception of early landraces and ancestors, mutations at two multiallelic loci, S and Z, mapped on chromosomes 1R and 2R, respectively, has led to self-fertility in the modern species. It has been proposed that self-fertility mutations have been captured in current hybrid ryes during breeding programs after introgression of a dominant self-fertility gene [50] from Iranian and Argentinean germplasms (Bolibok-Brągoszewska et al. [5] and references therein).

In DID, STR, MIN, MUS, CAR, PEV, and SUN, conversely, F values were negative, pointing to a condition of excess of observed heterozygosity. For SUN, this result was expected in consideration of its breeding history; current hybrids were obtained starting from genetically divergent inbreeds. In open-pollinated populations, experiencing a reduction of the effective size, an heterozygosis excess occurs, i.e., recently bottlenecked populations [51].

Under different circumstances, this can be the case in some of our rye accessions. For example, DID is repeatedly propagated at IPK to renew the ex situ accession, probably starting from a small number of seeds. In MIN, which is employed for the production of straw for packsaddles and only marginally for the grains, a condition of heterozygosity exceeding what is expected in a population at mutation drift equilibrium has most probably occurred.

Breeding lines in rye are thought to be exposed to severe inbreeding depression when repetitively self-fertilized due to the high load of recessive deleterious mutations accumulated in highly heterozygous species [52]. This seems not to be the case of CAR and PEV, which showed the lowest levels of inbreeding (F values = −0.259) among the alpine landraces.

Surprisingly, EST counterpoised the highest F value with the highest nucleotide diversity (π) recorded between accessions (Table 2, Figure 2a), meaning that even if a consistent proportion of alleles are homozygous, this landrace hosts a high number of SNPs in its genome. In this case, the π is definitely above the average, but for the other landraces affected by high F values we cannot report the same finding, revealing the presence of critical situations for conservation.

We included in our analysis an Italian accession of *S. strictum* Presl., which grows in southern Italy as a feral plant [53] and that could have potentially hybridized with several rye landraces [35,54]. *S. strictum* together with *S. vavilovii* are in fact wild ryes related to the cultivated winter rye. However, while many authors have questioned the validity of *S. vavilovii* as a separate species [55], others formulated the hypothesis that cultivated rye evolved from *S. strictum* [5]. The latter was chosen in this study because, differently from *S. vavilovii*, it occurs throughout the Mediterranean Basin [35,56] in perennial, open-pollinated populations [5] and it has been used in the past to create perennial rye hybrids [27]. However, the results of our study (Figure 1) have excluded recent hybridization phenomena of alpine landraces and varieties with this congeneric.

### 3.2. Genetic Structure

The structure of alpine rye populations turned out to be composed of three main genotypes, as observed in previous analyses using microsatellites and SNP array over a very large number of accessions [5], and of two main groups (cluster I and cluster II) (Figure 2b and Appendix A), similarly to the results of Targońska et al. [6]. We did not observe a clear geographic pattern in genetic structure among the observed landraces. At the local scale, it is difficult to retrace the presence of specific patterns, due the complex history of migrations and the intense commercial exchanges (Figure 2).

Since the two commercial varieties were included in cluster I, we further inspected the genetic structure of this group (DAPC and Prevosti’s distances; Figure 3 and Appendix A). Results revealed that many alpine landraces are tightly correlated with the commercial variety Antoninskie (ANT); for several of these landraces we can affirm that genetic distance between them and ANT was very small and probably they should be interpreted as a homogeneous genetic group (Appendix A).

The present study was conducted to identify valuable landraces from a sector of the Western Alps where the high rates of depopulation have involved a general abandoning of agriculture with the risk of strong crop genetic erosion. We found two genetic clusters that could represent two main lineages of ryes. Unfortunately, our data do not allow the full interpretation by an historical point of view, but considering the tight relation between cluster I with modern rye varieties, we can hypothesize that cluster II composed of MAR, MIN, ARN, and EST should consist of “ancestral” rye landraces which were grown in Alpine valleys since before the 50s of the twentieth century. This hypothesis is supported by high π values which characterized EST (0.003) together with MIN and MAR (where π is higher than the average value 0.00231), where genetic diversities are typical of many “traditional” landraces [12,57].

Divergent landraces of cluster II are similarly managed with family farming, despite coming from four rather distant sites. MIN was grown to be used as a filling for packsaddles and saddles. This rye has particularly short and thin stems. EST and MAR landraces showed high and thin stems, suitable for making thatched roofs, the traditional roofs from the SW-Alps villages. ARN comes from Arnad (Aosta Valley) in the NW-Alps, a peculiar area in the Italian landscape influenced by French culture, where thatched roofs were less common, but where rye stems were traditionally used for stable and barn roofs. An AFLPs analysis conducted by IAR and the University of Torino [58] on rye varieties from Aosta Valley, highlighted the presence of two main genetic clusters where ARN belonged to a genetic cluster characterized by distinctive morphological features such as precocity, green spikes, long and thin stems, small seeds and a global lower productivity. These features were found by us also in three landraces (data not shown; see Appendix A); our hypothesis is that the K1 cluster is composed of ryes which were principally selected as raw materials for buildings and artifacts, i.e., thatched roofs or in other manufactures [36,37,38].

Genetic distances between cluster I and cluster II were found to be well statistically explained by 39 loci, and eight of them were functionally assigned in previous studies (Table 4). Given the limits of this analysis, it is interesting that all genes at the most informative loci, were involved in pathways which characterize rye as a crop, such as: cold stress resistance, low amount of gluten proteins, major or minor ability to extend the stem, and so on. This is even more interesting if cluster I and cluster II are so different, as discussed above, and this aspect deserves further analysis with appropriate methods (i.e., QTL mapping and WGSs).

No apparent geographic structure was found among the alpine landraces of this study; a lack of correspondence between genotypes and the place of origin in rye has been documented [5,28,31]. Rye pollen is wind-dispersed for long distances [59] and seed accessions could be exchanged or shared by different farmers for several different reasons (commerce, emigration, etc.). Only in the case of BRU and DID landraces (belonging to cluster I; Figure 2) were we able to hypothesize a clear geographic correspondence between these two landraces coming from Bruzolo and San Didiero, which are very close localities in Valle Susa (TO). Both landraces were obtained from seed accessions stored at the IPK where, as already discussed, repetitive regeneration cycles could have affected genetic diversity [60,61]. Moreover, genetically distant landraces are very rare at a single geographical site [12,62].

Uniformity of the genetic pool of cluster I could be interpreted as a form of genetic contamination induced by random cross-pollination between different crops or depending on the farmers which deliberately mixed the different seeds. Alternatively, these landraces have originated from a common modern variety as Antoninskie itself or a very similar commercial variety.

### 3.3. Final Considerations

The presence of two clear genetic clusters within the W-Alps was likely to depend on the main local uses of these rye landraces; not only as foodstuffs, but also selected for roof building and the production of other manufacts. Landraces belonging to cluster II are rare, difficult to be obtained, and exposed to genetic erosion. The poor shelf life of rye seeds increases the vulnerability of this germplasm. On farm and ex situ preservation of the divergent landraces is encouraged, given their tendency to hybridize and the great wealth of genes they conceal.

It could be useful to remember that “authentic” local landraces are worthy of conservation, not only as part of a relict biodiversity, but also because landraces host genetic traits to respond to specific environmental stresses: traits which can be used in plant breeding and for the development of new resilient varieties.

As far as landraces of cluster I are concerned, a gene flow from commercial varieties towards them seems probable. Only four up to 16 landraces were completely separated, thus further investigation into their origin would be necessary. Morphological and agronomic analyses will clarify the relationship between some of these landraces and modern varieties. Most interesting landraces selected in this study are valuable under several agronomic aspects and they can be proposed in programs of crop enrichment and enforcement.

## 4. Materials and Methods

### 4.1. Plant Material Samples Preparation

In spring 2018, we started a survey to recover local rye varieties in NW-Italy. We recovered seeds of 10 rye varieties declared as “local” by the interviewed farmers. The Institut Agricole Régional (IAR) (Aosta, Italy), Ente Aree Protette Alpi Marittime (EAPAM) (Valdieri, Italy), and Liebniz Institute of Plant Genetics (IPK) (Leibniz, Germany) supplied seven landraces from Maritime, Graian, and Pennine Alps (Table 4, Figure 1). A commercial rye variety and a rye hybrid (ANT and SUN, respectively) widely cultivated in growing areas of North Europe and available also for the Italian market were used as controls. *S. strictum* seed accession was provided by IPK, supplying seeds from Italian Apennines, where the species grows wild.

Globally we analyzed 19 different seed accessions. Forty rye seeds (caryopsis) for each accession were sterilized for 60 s in 70% ethanol and then gently shaken twice in a sodium hypochlorite solution (10% commercial bleach in distilled water with 0.01% Tween20) for 5 min. Seeds were placed on wet sterile filter paper in petri dishes and incubated for two days at 4 °C and then moved to 24 °C with a 16/8 h light/dark cycle. After a few days (from one to seven), all mature seeds germinated. About 200 mg of fresh material (the first two leaves) for each sample were freeze-dried and stored at −20 °C. DNA extraction was performed with the NucleoSpin^®^ Plant II from Macherey-Nagel (Düren, Germany) as described in the product manual, using the PW2 buffer protocol. Samples’ quality was assessed by means of a NanoDrop^®^ ND-1000 spectrophotometer from Thermo Scientific (Waltham, MA, USA) respecting the ranges 260/280 ≥ 1.7 and 1.8 ≤ 260/230 ≤ 2.2. We quantified samples DNA concentration between 20 and 50 ng/μL, using a Qubit^®^ 2.0 fluorometer by Invitrogen (Carlsbad, CA, USA) using the DNA BR Assay Kit. Samples were shipped to IGA Technology Services (Udine, Italy) to be prepared and sequenced for ddRAD sequencing.

### 4.2. RADseq Genotyping

ddRAD (double digest Restriction Associated DNA) libraries were produced following the IGATech (Udine, Italy) custom protocol, with minor modifications with respect to Peterson et al. [63]. Genomic DNA, fluorimetrically quantified, was normalized to concentration and double digested with SphI and EcoRI enzymes. Fragmented DNA was purified with Agencourt AMPureXP beads (Beckman Coulter, Brea, CA, USA) and ligated to barcoded adapters. Samples were pooled on multiplexing batches and bead purified. For each pool, targeted fragments distribution was collected on a BluePippin instrument (Sage Science Inc., Beverly, MA, USA). Gel eluted fraction was amplified with oligo primers that introduce TruSeq indexes (Illumina adapters) and subsequently bead purified. The resulting libraries are checked with both Qubit 2.0 fluorimeter (Invitrogen, Waltham, CA, USA) and Bioanalyzer DNA (Agilent technologies, Santa Clara, CA, USA). Libraries were processed with Illumina cBot for cluster generation on the flowcell, following the manufacturer’s instructions and sequenced with V4 chemistry paired end 125bp mode on HiSeq2500 instrument (Illumina, San Diego, CA, USA).

### 4.3. Data Analysis

We analysed ddRAD-seq raw sequences using Stacks v2.0 [64] following the protocol described by Rochette and Cathchen [65]. We used the publicly available rye genome [66] to align RAD tags. We demultiplexed raw Illumina reads using the process_radtags utility included in Stacks v2.0. Sequences were aligned to the reference genome using BWA-MEM [67] with default parameters and selecting of uniquely aligned reads (i.e., reads with a mapping quality > 4). We detected all the covered loci from the aligned reads using the gstacks program and we filtered detected loci using the populations program in Stacks v2.0, which was ran with option *r* = 0.75 in order to retain only loci that are represented in at least the 75% of the population.

The populations.structure file produced by the populations program was used as input file in the following analysis performed in the R v3.6.2 environment [68] with the package poppr v2.8.6 [69].

The following steps were done to compare rye varieties’ genetic differentiation, excluding *S. strictum* samples (STR). We applied the clonecorrect function to identify duplicated multilocus genotypes into the original dataset and the informloci function to remove uninformative loci, setting the MAF threshold to 0.01. Expected and observed heterozygosity were calculated with the function poppr; the inbreeding index was calculated as F = (H_e_ − H_o_)/H_e_ [39]; and the nucleotide diversity (π) was calculated from the population program in Stacks environment [40,70]. Analysis of Molecular Variance (AMOVA) [71] was performed to estimate molecular variance among and between samples and populations, using the poppr.amova function and the randtest.amova function to verify the covariance components (1000 permutations). We characterized each population with several indices for heterozygosity, evenness, and linkage with the poppr function. Bayesian clustering, implemented in STRUCTURE v2.3 [72], was used to infer the number of putative genetic clusters. We ran the final simulations with 1 < K < 22 and 30 iterations for each K value; each run comprised a burn-in period of 10^5^ iterations, followed by 10^6^ Markov chain Monte Carlo steps. Following Evanno et al. [73], we defined the most adequate value for the number of clusters K with the online tool STRUCTURE HARVESTER [74]. A DAPC analysis [41] was performed with the function dapc to visualize relative genetic distances between individuals. Most informative loci were extracted using the loadingplot function using a restrictive threshold of 0.05; sequences were identified by BLAST searches and collected from GenBank (http://www.ncbi.nlm.nih.gov/genbank/, accessed on 10 August 2021), then functionally annotated for the locus function (when available).

We further inspected genetic distances within rye accessions calculating Prevosti’s distances among populations [75] with the prevost.dist function and visualizing them with a neighbour joining tree (nj). We sampled tree topologies with a 1000 bootstrap and we considered as significant a minimum bootstrap value of 75. We interpreted as variety-related landraces those falling under the rule “*Calibration of threshold levels for molecular characteristics against the minimum distance in traditional characteristics*” [76]; thus we applied the 0.02 genetic distance threshold as discriminant.

## Figures and Tables

**Figure 1 plants-10-02415-f001:**
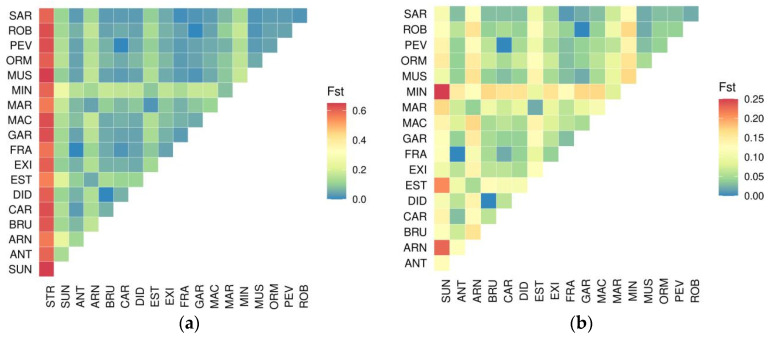
Pairwise genetic distances (Fst) between accessions. In both the two heat-maps, genetic distance is: Low distance 0.00 < x < 0.05 (in blue-green), Moderate distance 0.05 < x < 0.15 (in yellow-orange) and High to Very High distance 0.15 < x ≤ 0.25 (in dark orange-red). (**a**) Genetic distances are calculated for all the 19 accessions; (**b**) genetic distances are calculated after omitting STR (*S. strictum*).

**Figure 2 plants-10-02415-f002:**
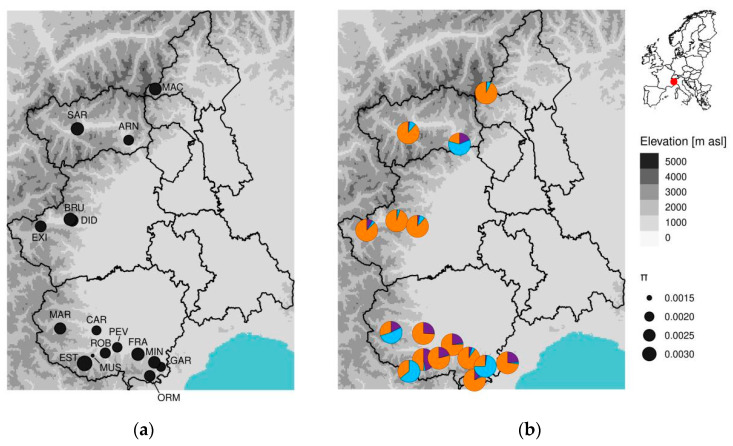
Geographic position of the rye landraces together with their nucleotide diversity and genetic structure. Each landrace is represented in the locality of collection along Western Alps. (**a**) Dot size corresponds to the mean nucleotide diversity. (**b**) Pie charts slices represent individuals probability to belong to one of the three genetic clusters calculated with STRUCTURE. Population IDs reported in (**a**) correspond to the populations in (**b**).

**Figure 3 plants-10-02415-f003:**
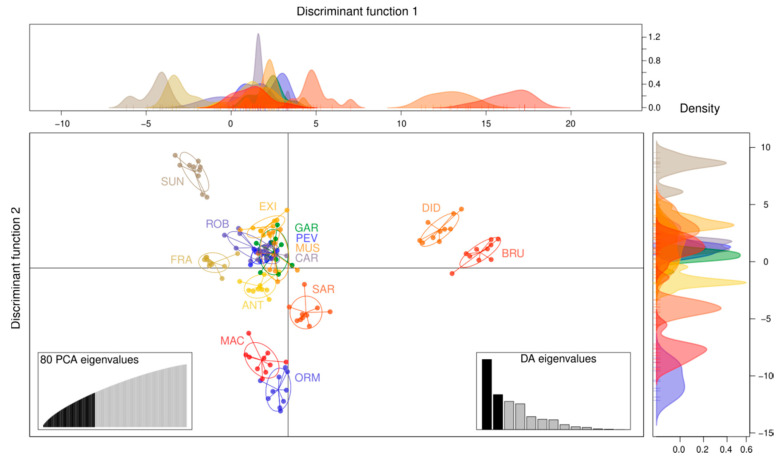
Relations between alpine landraces and modern varieties with a DAPC analysis. The multivariate analysis is based on 80 PCs. The majority of the landraces is related to Antoninskie (ANT). Density graphs (on the top and at the right of the DAPC representation) should help to interpret genetic distances between and within landraces and commercial varieties (ANT and SUN), and density corresponds to the genetic similarity within a single population.

**Table 1 plants-10-02415-t001:** AMOVA results. Sigma values representing the variance, σ, for each landraces/varieties (populations) and the percent of the total variance explained by each source of variance. Phi (ϕ) provides the landraces/varieties differentiation statistics. Higher Phi value corresponds to higher amount of differentiation. *p*-values refers to a two-tailed F-test.

Components of Covariance	Sigma	%	Phi	*p*-Value
Variance between populations (landraces/varieties)	170.5	29.8	0.30	<0.001
Variance between individuals within landraces/varieties	−31.7	−5.5	0.00	>0.05
Variance within individuals	432.7	75.7	0.24	<0.001
Total variations	571.5	100		

**Table 2 plants-10-02415-t002:** Estimates of genetic diversity per population. Observed (H_o_) and expected heterozygosity (H_e_) describe the proportion of heterozygous genotypes expected under the Hardy-Weinberg equilibrium [39] and the actual observed proportion of heterozygosity among samples (individuals) belonging to a population. F (inbreeding coefficient) is used to measure the inbreeding index in the population. Nucleotide diversity (π) is used to measure the degree of polymorphism within a population [40]. Accessions are ordered according to decreasing F values. Average values and standard deviation (SD) are reported at the bottom of the table for each genetic parameter.

Accession	H_o_	H_e_	F	π
EST	0.00281	0.00315	0.1079	0.00333
EXI	0.00197	0.00218	0.0963	0.00231
MAR	0.00206	0.00226	0.0885	0.00238
SAR	0.00235	0.00254	0.0748	0.00269
FRA	0.00236	0.00253	0.0672	0.00268
ANT	0.00205	0.00217	0.0553	0.0023
ROB	0.00193	0.00204	0.0539	0.00216
ORM	0.00204	0.00214	0.0467	0.00227
MAC	0.00232	0.00243	0.0453	0.00257
BRU	0.00238	0.00249	0.0442	0.00263
ARN	0.00192	0.002	0.04	0.00211
GAR	0.0018	0.00186	0.0323	0.00197
DID	0.00233	0.00232	−0.0043	0.00246
STR	0.00196	0.00192	−0.0208	0.00203
MIN	0.00241	0.00236	−0.0212	0.00249
MUS	0.00143	0.00138	−0.0362	0.00146
CAR	0.00214	0.00185	−0.1568	0.00196
PEV	0.00225	0.00194	−0.1598	0.00206
SUN	0.00238	0.00189	−0.2593	0.00199
Average	0.00215	0.00218	0.03	0.00231
SD	0.00025	0.00029	0.04	0.00031

**Table 3 plants-10-02415-t003:** Most informative loci. Best blast results for the most informative loci describing genetic distance between populations (DAPC-based).

Locus ID	Description	E Value	Id%	Accession	Putative Function	Reference
11747	*Triticum aestivum* cultivar Norstar CBFIVb-B9 gene, promoter region, 5’ UTR, and partial cds	2.00*e*-11	73.53%	EU562190.1	Cold acclimation	[42]
186075	*Triticum aestivum* cultivar Chinese Spring fructose-1,6-bisphosphate aldolase 19 (FBA19) gene, complete cds	4.00*e*28	93.41%	KY930464.1	Heat and cold stresses response	[43]
185806	*Triticum aestivum* gamma gliadin-A1, gamma gliadin-A3, gamma gliadin-A4, and LMW-A2 genes, complete cds	3.00*e*-24	95.71%	MG560140.1	Gluten protein	[44]
274554	*Aegilops tauschii* subsp. *tauschii* protein FAR1-RELATED SEQUENCE 9-like (LOC109763998), mRNA	5.00*e*92	81.07%	XM_020322871.1	Plant development regulation	[45]
499375	*Aegilops tauschii* clone BAC HD95L20 cytosolic acetyl-CoA carboxylase (Acc-2) and putative amino acid permeases genes, complete cds	6*e*-43	91.18%	EU660891.1	Flow of photosynth. C to metabolites	[46]
186075	*Aegilops tauschii* chromosome 1Ds prolamin gene locus, complete sequence	8*e*-28	93.41%	KY930464.1	Prolamine synthesis	[47]
255304	*Triticum turgidum* HMW-glutenin locus, complete sequence	1*e*-93	90.33%	AY368673.1	Gluten protein	[47]
276443	*Hordeum vulgare subsp. vulgare cultivar Nure frost resistance H2 gene locus, complete sequence*	6*e*24	93.88%	MN251600.1	Frost resistance	[48]

**Table 4 plants-10-02415-t004:** Rye Accessions list. For each seed accession, accession code, species, geographic provenance, elevation of the crop, life form, a brief description of the rye accession and main uses of the grains are given. Years of cultivation at the locality of sampling as declared by the owner (when available).

Accession ID	Species	Geographic Origin	Elevation ^1^	Life Form	Description	Years	Uses
STR	*S. strictum*	Monti-Sibillini, M. Rotondo, MC	-	perennial	Wild congeneric species supplied by IPK [R 898]	-	N.A.
SUN	*S. cereale*	N.A.	N.A.	annual	“Su Nasri” commercial hybrid, responsible of selection and conservation: Saaten-Union GmbH, Isernhagen Germany; supplied by RV Venturoli, Pianoro, Italy	-	EU Market, for food, feed and energy
ANT	*S. cereale*	N.A.	N.A.	annual	“Antoninskie” commercial open pollinated variety responsible of selection and conservation: Poznańska Hodowla Roślin Sp. z o.o., Tulce, Poland; supplied by Società Italiana Sementi, San Lazzaro di Savena, Italy	-	EU Market, for food, feed and energy
ARN	*S. cereale*	Arnad, AO	361	annual	Landrace supplied by IPK [R 1137]	-	N.A.
BRU	*S. cereale*	Bruzolo, TO	455	N.A.	Landrace supplied by IPK [R 1138]	-	Seed accession stored ex situ
CAR	*S. cereale*	Caraglio, CN	566	annual	Landrace	8–10	Own milling (French millstone), flour transformation and bread production for sale.
DID	*S. cereale*	San Didiero, (Fraz. Costa Pietra), TO	430	N.A.	Landrace supplied by IPK [R 1140]	-	Seed accession stored ex situ
EST	*S. cereale*	Entracque, CN	1150	annual	Landrace supplied by EAPAM	-	N.A.
EXI	*S. cereale*	Exilles (Locality Cels), TO	955	annual	Landrace, supplied by Molino Valsusa in Bruzolo (TO)	-	Own milling (millstones), flour and bread
FRA	*S. cereale*	Frabosa, CN	505	annual	Landrace	8	Own milling (millstones), flour, bread and biscuits for the family-run farm-house; bedding straw, thatched roofs
GAR	*S. cereale*	Garessio, CN	620	annual	Landrace	-	Flour and bread for the family-run farm-house
MAC	*S. cereale*	Macugnaga, VB	1327	annual	Landrace supplied by IPK [R 1136]	-	N.A.
MAR	*S. cereale*	Marmora (Locality Torello), CN	1410	annual	Landrace	20	Own milling (millstones) for a family-run farm-house
MIN	*S. cereale*	Garessio (Locality Mindino), CN	1000	annual	Landrace	-	Packsaddle and saddle straw production by the farmer
MUS	*S. cereale*	Sant’Anna di Valdieri, CN	1000	annual	Landrace, supplied by EAPAM, Ecomuseo della segale	-	A blend of grains established at the beginning of 2000 from several local landraces
ORM	*S. cereale*	Ormea, CN	1082	annual	Landrace	7	Flour and bread for family use
PEV	*S. cereale*	Peveragno, CN	515	annual	Landrace	10	Flour sale for catering
ROB	*S. cereale*	Robilante, CN	700	annual	Landrace supplied by M. Giordano in Robilante (CN)	40	Own milling (millstones) for a family use
SAR	*S. cereale*	Sarre-Bellon, AO	1400	N.A.	Landrace supplied by IAR	-	For family use as flour

^1^ meters above sea level; N.A. = Not Applicable; a dash was used to fill cells with unknown values.

## Data Availability

The data presented in this study are openly available in NCBI-SRA, BioProject ID: PRJNA765145.

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
