# Peer review of "A ddRADseq Survey of the Genetic Diversity of Rye (Secale cereale L.) Landraces from the Western Alps Reveals the Progressive Reduction of the Local Gene Pool"

_plants, 2021, doi:10.3390/plants10112415_

Round 1

Reviewer 1 Report

This study aimed to explore the genetic diversity of rye landraces in Italian NW-Alps based on the SNPs data obtained from RAD-seq. The extent of genetic diversity and population differentiation were compared among the landraces together with wild congeneric species (S. strictum), commercial rye variety (ANT) and the rye hybrid (SUN). The results showed that the majority of genetic diversity was present between landraces and varieties. Many landraces were genetically close to the commercial variety ANT, while several landraces were relatively distant form ANT. Considering a preservation of landraces for improving the rye varieties, it is important to explore genetic diversity in the landraces stored by local farmers. In this regard, this study reveals successfully that a few genetically distinct landraces are still maintaining by family farming.

The manuscript is generally well-written and understandable enough, and the methodologies seems to be relevant. I only have several specific comments and show them below.

The STRUCTURE analysis was conducted with number of clusters (K) between 1 and 22 (line 467), but I am wondering why Ln’(K), |Ln”(K)| and Delta K for K = 2 are not available in Table S1. 

"The larger group (cluster II) contained most of the rye landraces and the two modern varieties, SUN and ANT; the four landraces ARN, EST, MIN and MAR grouped apart (cluster I) (lines 166-168)." The larger group should be cluster I and the four landraces be cluster II as shown by Figure S1.

At lines 301-306, is it possible that higher nucleotide diversity with high inbreeding coefficient is due to the presence of two genetically isolated populations in the landrace (the Wahlund's principle)?

In Table 4, there are 19 accession ID in total. Three of them are congeneric species (STR), or commercial varieties (SUN and ANT), and the remaining 16 are landrace. Although I am afraid of my misinterpretation, in line 391 authors describe “Only three up to 15 landraces”. Would it be 16 landraces? Besides that, would three landraces be four? This is because in figure S2, for landraces (MIN, MAR, ARN, EST) are in cluster II. I am also wondering why MIN is not included in Figure S3. Authors should solve these conflicts.

At line 132, S. strictum -> intalic

At line 310, Authors -> authors

At line 311, S. vavilovii -> intalic

Author Response

Dear reviewer, thank you for the useful comments and suggestions. Following our replies to the comments:

- The STRUCTURE analysis was conducted with number of clusters (K) between 1 and 22 (line 467), but I am wondering why Ln’(K), |Ln”(K)| and Delta K for K = 2 are not available in Table S1.

> We added the K=1 and the missing K=2 values.

- "The larger group (cluster II) contained most of the rye landraces and the two modern varieties, SUN and ANT; the four landraces ARN, EST, MIN and MAR grouped apart (cluster I) (lines 166-168)." The larger group should be cluster I and the four landraces be cluster II as shown by Figure S1.

> This is true, we corrected the sentence.

- At lines 301-306, is it possible that higher nucleotide diversity with high inbreeding coefficient is due to the presence of two genetically isolated populations in the landrace (the Wahlund's principle)?

> The Wahlund effect is certainly an interesting paradox, but it is usually associated to the low heterogeneity, since the nucleotide diversity in EST population is high, we can exclude the presence of this effect.

- In Table 4, there are 19 accession ID in total. Three of them are congeneric species (STR), or commercial varieties (SUN and ANT), and the remaining 16 are landrace. Although I am afraid of my misinterpretation, in line 391 authors describe “Only three up to 15 landraces”. Would it be 16 landraces? Besides that, would three landraces be four?

> We changed the sentence in “Only four up to 16 landraces”.

This is because in figure S2, for landraces (MIN, MAR, ARN, EST) are in cluster II. I am also wondering why MIN is not included in Figure S3. Authors should solve these conflicts.

> Figure S3 is focused on populations from cluster I, thus MIN was not included in the analysis.

- At line 132, S. strictum -> intalic

- At line 310, Authors -> authors

- At line 311, S. vavilovii -> intalic

> Done

Reviewer 2 Report

Review of the paper titled: „A ddRADseq survey of the genetic diversity of rye (Secale cereale L.) landraces from the Western Alps reveals the progressive reduction of the local gene pool”      written by: Martino Adamo, Massimo Blandino, Luca Capo, Simone R. Enri, Anna Fusconi, Michele Lonati, Marco Mucciarelli

Dear Editor,

I had a pleasure to read and rewie of the paper (title above) that concern on an interesting subiect as Rise an his landraces in Europe. Authors has written an interesting paper concerning  ex situ conservation of genetically divergent landraces that Has been which was preceded by advanced genetic tests. The aim of this study was to explore the genetic diversity in Italian Alps Secale cerale landraces, in order to understand which of them deserve conservation efforts,due to its “authentic” adaptation to local environmental conditions.  Below I am addin some critical facts on this paper.

Introduction

The introduction is generally well written and interesting. Lots of information about the secale cerale varieties and the importance of this grain. In the introduction, the authors briefly describe the history of the affirmation and urawa Secale cerale and recent initiatives that started to re-evaluate the importance of this crop local varieties and to preserve their diversity by ex-situ and on-farm conservation

Except of recently a complete genome of rye was assembler, also several studies on relationships within and between Secale species have been carried out to study the genetic variability, using a variety of different molecular methods. In general, the aim of rewieved study was to explore the genetic diversity in still viable Italian Alps rye landraces, in order to understand which of them deserve conservation efforts, due to its “authentic” adaptation to local environmental conditions. I believe that the conclusions drawn from this article, contained in the article and based on reliable research, are valuable and worth publishing. The remaining comments are contained in the remaining parts of the review. I strongly agree with the research goals set by the authors.

Results

The results obtained during the tests are well described and reliably presented, and the calculated parameters are reliable. It is usually a good idea to use Structure-based analyzes or similar in similar work. The results obtained in this way are a kind of standard, although of course such calculations are not replaced. Using the Structure program, the authors presented very vividly the diversity of the studied populations.

Discussion

The discussion is written correctly with a large number of cited publications, which indicates familiarity with the topic that the authors analyzed. In the discussion and summary, the authors actually re-summarize the results regarding the possibility of hybridization of local Secale cerale breeds. The summary lacks some information on the practical application of the obtained results and how the results could be directly implemented in the agricultural industry. There is definitely no information as to the application of the selected material and whether the material was actually used or can be used to enrich new varieties with new features that occur in older Alpine varieties. The discussion also includes a fairly well-known and stated fact that no apparent geographic structure was found among the landraces (in this case alpine). Rye pollen is wind-dispersed for long distances - that is widely known. The discovery should also include the broader conclusions drawn from Fig. 2, on which pie charts slices represent individuals probability to belong to one of the three genetic clusters. Sharply to the north, towards the Alps, the variability and genetic diversity of the Secale cerale population are decreasing. Please explain what, according to the authors, is the cause of this? And did the migrations of people, which undoubtedly occurred thousands of years ago, have an influence on the greater genetic variability of the populations south of the Alps? What about populations even more geographically distant from the Alpine populations? In addition, maybe it is worth adding what the variability of Secale cerale looks like in a wider geographical range and what conclusions can be drawn from this not only for the cultivation but generally for the biology of other grasses? Maybe it is also worth adding why some landraces can be considered as a single genetic unit and why is this so?

Despite these few criticisms, I strongly believe the manuscript is worth publishing in Plants. The above criticisms do not in any way reduce the quality of the article.

Author Response

Dear reviewer,

thank you for the very positive comments to this study, we really appreciated the numerous comments and suggestions to improve our work. In the following paragraph you can find a brief reply to any raised point.

- In the discussion and summary, the authors actually re-summarize the results regarding the possibility of hybridization of local Secale cerale breeds. The summary lacks some information on the practical application of the obtained results and how the results could be directly implemented in the agricultural industry.

> We add the following sentence in the conclusion paragraph: “It could be useful to remember that “authentic” local landraces are worthy of conservation, not only as part of a relict biodiversity, but also because landraces host genetic traits to respond to specific environmental stresses: traits which can be used in plant breeding and for the development of new resilient varieties.”

- There is definitely no information as to the application of the selected material and whether the material was actually used or can be used to enrich new varieties with new features that occur in older Alpine varieties.

> Currently selected varieties are not included in breeding programs, nevertheless we are writing a paper on their agricultural and nutritional traits (unpublished data), thus we do not exclude a future exploitation in this regard. We already cited the opportunity to perform trait-based studies on the selected landraces in rows 336-338.

We also added a “perspectives” paragraph, at the end of the paper, in order to clarify this point: “Most interesting landraces selected in this study are valuable under several agronomic aspects and they can be proposed in programs of crop enrichment and enforcement.”

- The discovery should also include the broader conclusions drawn from Fig. 2, on which pie charts slices represent individuals probability to belong to one of the three genetic clusters.

> We added a comment referred to Figure 2 (rows 299-302): “We do not observed a clear geographic pattern in genetic structure among the observed landraces. At local scale it is difficult to retrace the presence of specific patterns, due the complex history of migrations and the intense commercial exchanges (Figure 2).”

- Sharply to the north, towards the Alps, the variability and genetic diversity of the Secale cerale population are decreasing. Please explain what, according to the authors, is the cause of this?

> As far as we have observed, we have not detected a latitudinal gradient in genetic diversity or variability. As regards to the number of landraces analysed, many come from the Ligurian and Maritime Alps, fewer from the northernmost sectors of the Alps. This effect has created a sampling bias, in fact, in the Ligurian and Maritime Alps there was a greater willingness of farmers to share the grain they had, if compared to the other parts of the study area.

- And did the migrations of people, which undoubtedly occurred thousands of years ago, have an influence on the greater genetic variability of the populations south of the Alps?

> Our analyses focused on the concept of landrace, this particular type of unit, can originate even in a rather short period of time. Some agronomists consider about seven years the time enough to create a new landrace. It is therefore difficult to associate the genetic variability of landraces with specific population migrations, especially if it occurred in a local context, where trade can greatly complicate the tracing of their genetic origin. In addition to these observations, we must admit that information on the origin of landraces was often incomplete, if not missing, which makes it even more difficult to place landraces in a historical context.

- What about populations even more geographically distant from the Alpine populations?

> We discussed the argument in rows 231-244, we found particularly interesting two extensive studies (in ref. 5 and 35).

- In addition, maybe it is worth adding what the variability of Secale cerale looks like in a wider geographical range and what conclusions can be drawn from this not only for the cultivation but generally for the biology of other grasses?

> We are conscious that it is worth to compare our landraces with those from a wider geographical range, however the ddRAD sequencing sampling depth is difficult to compare with studies based on less informative methods. We are confident that conservation of rye landrace from a limited geographic area, as in the case of Ligurian and Maritime Alps, can as efficient as conserving landraces from a wider geographic range [see ref. 35].

Crop genetics are strongly influenced by the different breeding systems, thus we choose not to speculate over our conclusions extending some considerations to the grasses biology in general.

- Maybe it is also worth adding why some landraces can be considered as a single genetic unit and why is this so?

> This is a very good observation, since it is almost a philosophical question. To define a single genetic unit we used a conservative approach. We based it on the genetic distance of what is currently known as two different commercial varieties (in our case Antoninskie and Su Nasri). We explained the concept at the end of material and methods (rows 458-464; see Figure S3).